# Investigation on Performance and Kerf Characteristics during Cryogenic-Assisted Suspension-Type Abrasive Water Jet Machining of Acrylonitrile Butadiene Rubber

Preeti Maurya, Gaddale Srinivas Vijay and Raghavendra Cholpadi Kamath *

Department of Mechanical and Industrial Engineering, Manipal Institute of Technology,
Manipal Academy of Higher Education, Manipal 576104, Karnataka, India
* Correspondence: cr.kamath@manipal.edu

**Abstract:** The need for soft polymer (such as acrylonitrile butadiene rubber (ABR)) components in mating applications is increasing in several sectors, viz. automobile, mining, and marine, due to their viscoelastic nature with improved surface quality and tighter geometric tolerances. Therefore, this paper aims to compare the effect of cryogenic conditions on the performance parameters of the suspension-type abrasive water jet (S-AWJ) machining and investigate the kerf characteristics of the top and bottom surface by comparing the waviness of the cut profiles and abrasive contamination of the top surface near the vicinity of the slot under conventional (room temperature) and cryogenic (liquid nitrogen ($LN_2$)) conditions. The study found that the use of $LN_2$ positively affected the performance parameters (Kerf taper ratio ($KT_R$) and material removal rate (*MRR*)) due to a sudden increase in Young's modulus and a decrease in elasticity of the machining zone. The cryogenic-assisted S-AWJ at the highest water jet pressure (*WJP*) (250 bar) produced better kerf characteristics through uniform and waviness-free top and bottom kerf profiles than the other experimental sequences. The use of $LN_2$ resulted in the embrittlement of ABR, due to which less garnet abrasive particle contamination was observed during cryogenic-assisted S-AWJ machining.

**Keywords:** machining; abrasives; rubber; hydraulic jets; cryogenic; kerf; contamination; taper

## 1. Introduction

The abrasive water jet (AWJ) is the only cold high-energy beam technology with several distinct processing advantages for machining various materials [1,2]. AWJ performs better while machining viscoelastic materials, such as rubber [3,4]. The AWJ is classified into two categories, viz., Injection-type jet and Suspension-type jet. The suspension-type abrasive water jet (S-AWJ) machining method produces precision components for automobile, mining, and marine applications. Hollinger introduced a novel water jet method in 1989 that uses a suspension of abrasive particles and a high-polymer solution [5]. The S-AWJ has significant potential in the machining field due to its high coherence and energy efficiency benefits. Previous researchers have reported on the S-AWJ machining of various materials, but little research has been carried out on acrylonitrile butadiene rubber (ABR).

The machining of ABR is challenging compared to other engineering materials due to its viscoelastic nature. It is not easy to hold ABR during machining. As per the previous researchers, there are two methods for improving the machinability of viscoelastic materials. The first is the rising deformation rate, which results in a stiffer material with a higher storage modulus (G') and loss modulus (G''), which define the elastic and viscous behavior of the elastomers, respectively. Another method is cooling viscoelastic material using cryogenics during machining. While comparing both methods, cooling ABR is far more effective, since the temperature impacts elastic modulus more than strain rate. The cryogenic environment allows the ABR to withstand high applied forces, but it can fracture

because it cannot bend easily. As per the latest review on the machining of viscoelastic material under cryogenic conditions, conducted by Maurya et al. [6], the structural properties change and Young's modulus of these materials increases, which help in the improvement of the erosion rate, cutting force, surface morphology, chip formation, and reduction in the abrasive particle embedding at the machined surface. Apart from the material perspective, cryogenic machining is eco-friendly, economical (due to multiple uses of the same cryogenic setup), has no environmental side effects, and there is less possibility of a heat-affected zone in the machining area. Due to this, Nayak et al. [7] worked on turning ABR using a conventional machining method (lathe machine) under cryogenic conditions (Solid carbon dioxide). The authors found that cryogenic conditions led to the easy flow of chips over the tool surface, continuous chip formation, and reduced tool wear. The authors considered the limited output parameters: machining force, radial force, feed force, and chip morphology. Apart from the conventional machining method, Maurya et al. [3] recently attempted S-AWJ machining of the thick ABR workpieces (10 mm) used in mining equipment at room temperature conditions. The authors noticed an absence of cracks and wavy edges near the machined kerf profiles. The authors recommended that further investigation is required to determine the optimum values of the process parameters to achieve the waviness-free kerf profile and analyze abrasive contamination near the machined kerf profile. However, previous researchers have not revealed kerf characteristics studies in the S-AWJ-machined ABR surface.

The surface is one of the vital features in the machining process since it determines the efficient functioning of the machined components [8]. Due to the necessity of subjecting the components to mating applications or wear and friction conditions, the demand for enhanced surface quality and tighter geometric tolerances in machined ABR gaskets and bushes has increased in the positive displacement motors used in the mining sector [9]. The use of diverse manufacturing processes, including mechanical, metallurgical, chemical, thermal, and electrical, has modified the manufacturing of the component's surface due to the machining process's inherent features. Previous studies have found that the AWJ method improves the kerf characteristics due to less heat generation and burr formation in the machining zone [3,4]. Some previous researchers [10–12] have worked on the kerf characteristics examination of components machined using AWJ, and the authors asserted that the functionality of the manufactured components improved. During the experimentation, the authors incorporated the kerf width, kerf taper, kerf geometry, and abrasive contamination analysis in the kerf characteristics study. Several researchers studied the kerf characteristics of the machined slot on different materials using AWJ machining [10–14]. Kumaran et al. [15] found that edge delamination and corner deformation are the common issues during the AWJ machining of carbon-fiber-reinforced polymer, as shown in Figure 1. The latest research [10] obtained similar results during the AWJ machining of the carbon-fiber-reinforced polymer. The authors observed edge rounding and a wavy slit at the machined top kerf profiles. On the other hand, the bottom kerf profile included slits, wavy edges, incomplete penetration, and increased width at jet endpoints. The kerf characteristics study of the current work focuses on analyzing the geometry of the slot at the top and bottom surfaces and abrasive contamination at the top surface near the vicinity of the machined profile under conventional and cryogenic conditions.

The upcoming detail includes the finding of the previous researchers, which deals with the effect of process parameters on the performance parameters of AWJ-machined surfaces under conventional and cryogenic conditions. Mardi et al. [16] have reported that an increase in traverse rate results in grooving, irregularity, and a rise in surface roughness in the AWJ-machined surface of metal matrix composite. The primary reason for irregularity and surface roughness was the increased traverse rate of the AWJ nozzle. As per the latest review by Liao et al. [17], the AWJ machining of the metal matrix composites was formed by pronounced machining traces of various lengths and widths left by abrasive grains. The major surface defects were plastic deformations, cracking pits and voids, abrasive contamination, and micro-melting. Salinas et al. [18] studied the surface roughness,

topography, depth of cut, and residual stress during the AWJ milling of Inconel 718 and observed that water jet pressure affected surface roughness and depth of cut significantly. In contrast, the traverse rate and stand-off distance significantly affected the residual stress. Ramakrishnan [19] looked at the surface roughness and microhardness of the titanium alloy in three distinct regions (i.e., initial damage zone, smooth machining zone, and rough machining zone) produced by AWJ machining. The authors claimed that jet pressure was the most influencing factor in reducing surface roughness and decreasing waviness on the AWJ-machined surface. Yuvaraj and Kumar [11] investigated the surface topography and roughness profile in the cryogenic-assisted and conventional AWJ machining of aluminum alloys and reported that AWJ machining maintains surface characteristics by using abrasives with medium and fine mesh sizes along with oblique impingement angles of a jet. Furthermore, surface characteristics were improved during the cryogenic-assisted AWJ machining of aluminum alloys [11].

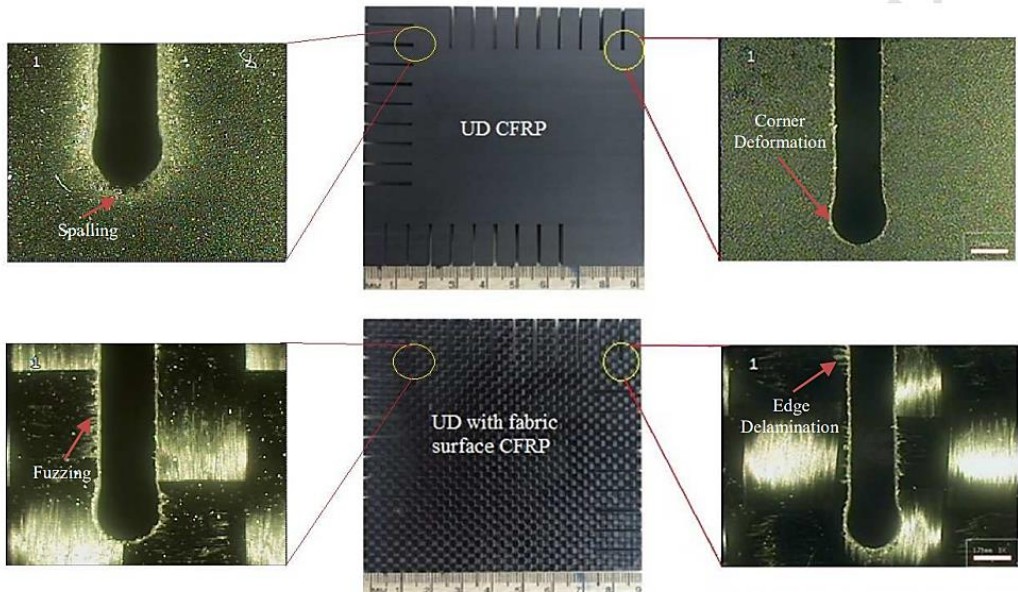

**Figure 1.** Edge delamination and corner deformation images (reproduced with permission) [15] of the machined carbon fiber reinforced polymer using AWJ machining.

Besides the studies conducted on the AWJ machining of different engineering materials, viscoelastic material (soft polymer) machining has been of great interest due to its growing application in the automobile, biomedical, and mining sectors. Very few studies have focused on the S-AWJ machining of these materials. Kowsari et al. [20] reported that S-AWJ machining on the polymethylmethacrylate produced a smoother surface than the as-received surface. The authors concluded that ductile plastic deformation was the dominant erosion mode to achieve a smoother surface during the machining of soft polymers. Tamannaee et al. [21] developed a model to predict the depth and waviness of the machined surface in the talc-filled thermoplastic olefin using repeated passes of the S-AWJ nozzle. It was shown that adding shallow smoothing passes (single pass/two passes) could prevent waviness. It may be noted that except for Tamannaee et al. [21], no other attempt is made by any of the researchers to investigate the kerf characteristics of the S-AWJ machining of the soft polymer. In that regard, the outcome of the present research will be significant, as it not only aims to check the effect of cryogenic conditions on the performance parameters of the S-AWJ machining of ABR but also investigates the kerf characteristics by comparing the waviness and abrasive contamination under conventional and cryogenic conditions.

## 2. Materials and Methods

The section includes the details of the experimental test setup, work material, machining process/performance parameters, and surface morphology studies used.

### 2.1. Cryogenic-Assisted S-AWJ Test Setup, Test Procedure, and Work Material

The cryogenic setup attached to the S-AWJ machine is shown in Figure 2a. The experiments were carried out on the custom-made two-axis CNC S-AWJ machine setup with a maximum pressure of 60 MPa. The specification of the S-AWJ machine is shown in Table 1. The slot machining was performed under conventional (room temperature) and cryogenic (liquid nitrogen ($LN_2$)) conditions on the selected work material, i.e., ABR. The 20 mm slot length was machined on each workpiece (refer to Figure 2b) under both conditions. The S-AWJ machine used the high-pressure slurry jet (suspension mixture), which is a mixture of water, polymer, and abrasive, to produce a slot on the ABR. In a suspension mixture, the Zycoprint polymer works as a thickener and enhances the viscosity of the mixture. It is a copolymer of an ammonium salt and other ingredients, such as surfactants and paraffin oils. The 1% concentration of the Zycoprint polymer solution has a higher viscosity than the commercial polymer solution at 1.5% [22]. The choice of the nozzle is one of the most crucial elements in AWJ machining. Silicon carbide, tungsten carbide, boron carbide, composite carbide, etc., are often-used materials to manufacture nozzles. Each material has a unique characteristic that identifies the capacity of the nozzle to extend its life cycle. The tungsten carbide has the highest toughness in abrasion compared to the other materials, as observed by the previous researcher [23]. Because of this, the customized stainless-steel tungsten carbide (SSTC) nozzle of 1 mm orifice diameter (shown in Figures 3a and 4a) was developed to direct the suspension mixture onto the surface of ABR. Under conventional conditions, the ABR was machined by supplying the high-pressure slurry jet at room temperature, as shown in Figure 3. On the other hand, under cryogenic conditions, the continuous flow of $LN_2$ was provided at the top surface of ABR during machining with a high-pressure slurry jet, as shown in Figure 4. The $LN_2$ was delivered at the top surface of ABR by a stainless-steel nozzle via an insulated stainless-steel braided hose pipe using compressed air at an angle of 80° and temperature of −196 °C during the cryogenic-assisted S-AWJ machining. The air compressor with a maximum pressure of 12 bar is attached to the $LN_2$ container (TA-55) to transfer the compressed air, as shown in Figure 2c. As per the previous researchers [24–26], the selection of $LN_2$ inlet pressure is critical, as it affects the outlet condition of $LN_2$. In the current study, $LN_2$ container pressure of 0.6 bar is selected for the cryogenic experiments based on the literature survey [26,27] and trial tests to avoid freezing of the slurry near the machining zone. In order to control the inlet pressure in the $LN_2$ container, the pressure regulator was attached before delivering compressed air, as shown in Figure 2a. Cryogenic cooling transforms the ABR from its elastic to tough phase, which results in an increase in modulus and a decrease in elongation. Hence, it improves the machinability and kerf characteristics of the ABR during cryogenic machining. The dimensions of the workpieces selected for both conditions were different, considering the machine worktable constraint and minimizing the material cost. The rectangular cross-section with a size of 50 mm × 80 mm and a thickness of 15 mm was chosen for conventional conditions. During machining, the spacing between two slots was set as 10.00 mm to avoid the influence of kerf characteristics on the adjacent kerf. This allowed the machining of six slots on one workpiece for conventional conditions. Under cryogenic conditions, since the workpiece was exposed to the $LN_2$ flow during machining, only one slot per workpiece was produced. Thus, the workpiece dimension considered for cryogenic conditions was 50 mm × 40 mm × 15 mm.

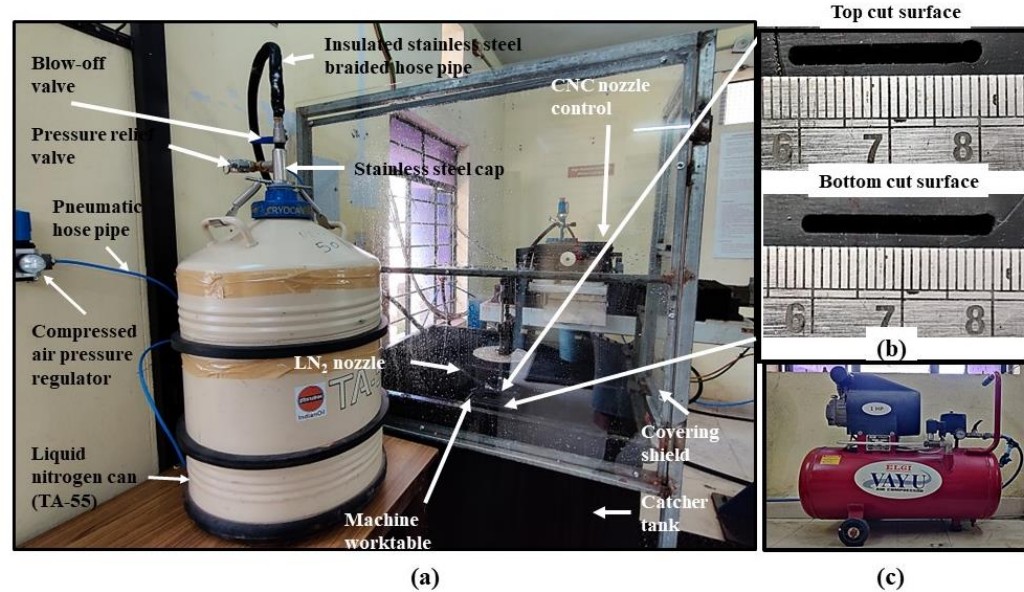

**Figure 2.** (**a**) Detail of cryogenic setup attached to the S-AWJ machine, (**b**) photographic images of the top and bottom cut surface on the ABR workpiece, and (**c**) air compressor.

**Table 1.** Specification of S-AWJ machine.

| Item | Description |
|---|---|
| Hydraulic plunger pump | Triplex reciprocating pump<br>Direct driven<br>Power = 40 HP<br>Discharge = 16 ltr/min<br>Delivery Pressure = 60 MPa |
| Floating piston cylinder | Capacity = 10 ltr |
| Suspension charging tank | Capacity = 50 ltr |
| Suspension mixing tank | Capacity = 100 ltr |
| Air compressor | Power = 1 HP<br>Capacity = 45 ltr<br>Delivery pressure = 12 bar |
| CNC nozzle movement | Two axis control<br>Maximum travel in X-direction = 450 mm<br>Maximum travel in Y-direction = 450 mm |

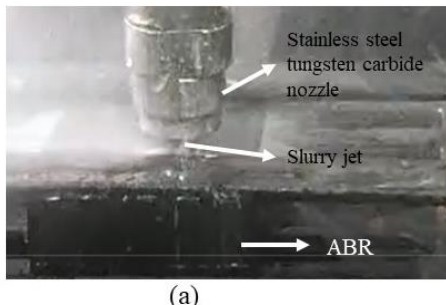
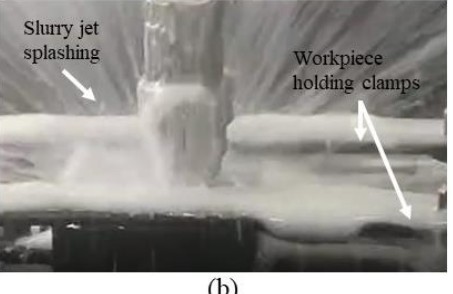
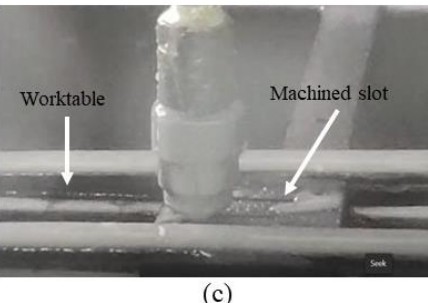

**Figure 3.** Photographic image of (**a**) jet initialization, (**b**) jet penetration, and (**c**) end of the machining of ABR workpiece using high-pressure slurry jet under conventional condition.

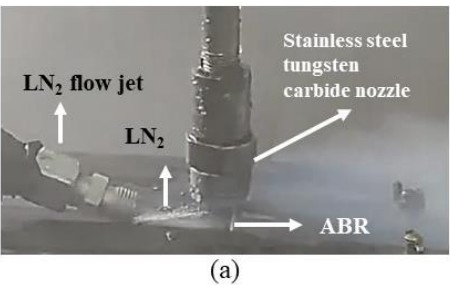 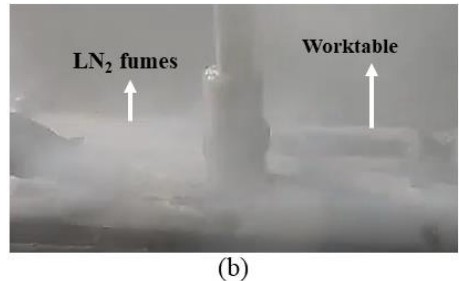 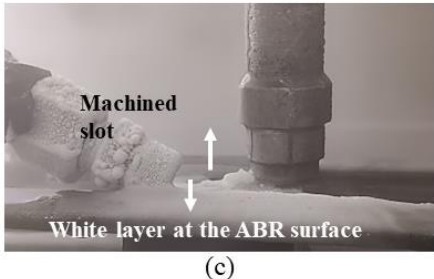

**Figure 4.** Photographic image of (**a**) jet initialization/LN$_2$ flow, (**b**) jet penetration/LN$_2$ fumes, and (**c**) end of the machining of ABR workpiece using high-pressure slurry jet under cryogenic condition.

### 2.2. Selection of Machining Process Parameters, Experiment Test Sequence and Performance Parameters

A Scanning electron microscope (SEM) (make: ZEISS, USA, model: EVO MA18) was used to analyze the shape factor of the garnet particles (by Equation (1) [28]) to achieve the shape irregularity before machining. The low shape factors (e.g., 0.4) correspond to an irregular/angular-shaped grit, whereas high shape factors (e.g., 1.0) indicate a more round-shaped grit [29].

$$F_{shape} = \frac{d_{min}}{d_{max}} \tag{1}$$

where, $d_{min}$ and $d_{max}$ are the minimum and maximum diameters of the abrasive particles. The statistical determination of the shape factor of garnet abrasive particles was carried out by analyzing the SEM image using digital image analysis software (image processing and analysis in Java (ImageJ), rsbweb.nih.gov/ij/, (accessed on 1 August 2022)) to find out the range of shape factors having maximum percentage quantity. Along with SEM, the energy dispersive X-ray (EDX) spectroscopy was conducted on garnet and ABR to determine the mineral composition weight percentage before machining.

The machining process parameters and their levels are summarized in Table 2. A jet impingement angle of 90° was the fixed process parameter because it can narrow down the kerf width to 164.9% of the nozzle diameter when SOD is ≈1 mm, as per the latest research [30]. The main reason behind selecting two machining passes was to produce a waviness-free and better kerf profile on the workpiece. The garnet abrasive with 80 mesh size is a commonly used low-cost abrasive in the S-AWJ machining [31] and was selected for the present research work. In addition, AWJ nozzle wear is less while using the garnet abrasives than other commercially available abrasives [23,32]. The reason behind selecting the stainless-steel nozzle with a tungsten carbide orifice is discussed in Section 2.1. The selected orifice diameter for the current work is 1 mm, which can be attributed to the good quality and efficiency in machining with AWJs, as suggested by the previous researcher [33]. The reasons behind selecting fixed process parameters related to cryogenic setup, viz., LN$_2$ flow pressure and angle, are discussed in Section 2.1. The mass percentage of Zycoprint polymer ($\omega_p$) and garnet ($\omega_a$) were considered to be 0.8% and 3%, respectively, as per the recommendation given by previous researchers [34]. It is possible to achieve a suspension mixture free from garnet sedimentation at the recommended percentages. The trial tests were conducted on the slurry to achieve a stable mixture. Apart from the fixed process parameters, the water jet pressure (*WJP*), traverse rate ($V_f$), and stand-off distance (*SOD*) were considered variable process parameters. On the basis of the literature [3,7,11,13,14,16,24,30], these dominant variable process parameters were chosen. The ranges of S-AWJ machining parameters selected for the experimentation were tested in preliminary machining trials. The experimental test sequences were constructed using Taguchi's L$_{27}$ orthogonal array, which allows for a smaller number of runs needed than the conventional central composite design used along with response surface methodology. Hence, the use of Taguchi's method resulted in saving resources, expenses, and time.

**Table 2.** The cryogenic-assisted S-AWJ machining process parameters.

| Process Parameters | |
| --- | --- |
| **Fixed** | **Variable** |
| Jet impingement angle: 90° | Water jet pressure (*WJP*): 150, 200 and 250 bar |
| Number of passes: 2 | Traverse rate ($V_f$): 40, 50 and 60 mm/min |
| Abrasive: garnet | Stand-off distance (*SOD*): 1, 1.5 and 2 mm |
| Abrasive mesh size: #80 | |
| Focusing nozzle: Stainless steel | |
| Orifice: tungsten carbide | |
| Nozzle orifice diameter: 1 mm | |
| $LN_2$ flow pressure: 0.6 bar | |
| $LN_2$ flow angle: 80° (with nozzle axis) | |
| $LN_2$ flow rate: 1.5 lpm | |
| Mass percentage of Zycoprint polymer ($\omega_p$): 0.8% | |
| Mass percentage of garnet ($\omega_a$): 3% | |

The Kerf taper ratio ($KT_R$) and material removal rate (*MRR*) were selected performance parameters. The $KT_R$ is calculated using Equation (2). A low $KT_R$ ($\approx 1$) is desirable for a better-quality cut.

$$KT_R = \frac{KT_T}{KT_B} \qquad (2)$$

where $KW_T$ and $KW_B$ are top and bottom kerf width, measured using a toolmaker's microscope (Model: TM-505B, make: Mitutoyo, least count: 0.005 mm and magnification of 10×) and optical microscope (Model: DP 22, make: Olympus, magnification of 50×). The five different readings from $KW_T$ and $KW_B$ were taken on each machined slot using each instrument, and the average readings were calculated to minimize the error. The *MRR* is measured using Equation (3), representing the volume of the material removed by the S-AWJ in unit time. A high *MRR* is desirable for a better-quality cut.

$$MRR \left( \frac{mm^3}{min} \right) = t \times w \times V_f \qquad (3)$$

where, $t$ = thickness of the workpiece (15 mm), $w$ = width of the kerf (mm) = ($KW_T$ + $KW_B$)/2 and $V_f$ = traverse rate of S-AWJ nozzle (mm/min).

### 2.3. Surface Morphology

The kerf characteristics of the machined slot ($KW_T$ and $KW_B$ profiles) were observed under conventional and cryogenic conditions using a profile projector vision plus (make: Metzer, model: Metz-300 T.T Supreme) (refer to Figure 5a) and optical microscope (refer to Figure 5b). The images were taken using a mobile (Model: OnePlus 9R) with a 48-megapixel camera in UltraShot HDR mode with a magnification of 2.5× by maintaining a constant distance from the profile projector. The kerf profile images of the $KW_T$ and $KW_B$ were taken to check the uniformity in the geometry of the machined slot. The analysis was performed by neglecting the 5 mm length from both ends of the machined slot (10 mm removed from a total of 20 mm slot length) to exclude the initial damage zone and to obtain a justified comparison between the kerf profile produced under conventional and cryogenic conditions by considering the smooth machining zone.

The SEM and EDX analysis were performed on the workpieces before and after machining to compare the number of garnet particles embedded on the top surface at the vicinity of the slot. Since it was more challenging to visually identify embedded particles on the machined surface, the SEM images were captured with the backscatter electron (BSE) detector to reveal the embedded abrasive particles on the top surface of ABR under conventional and cryogenic conditions. Then, the images were analyzed using ImageJ software to identify the embedded particles.

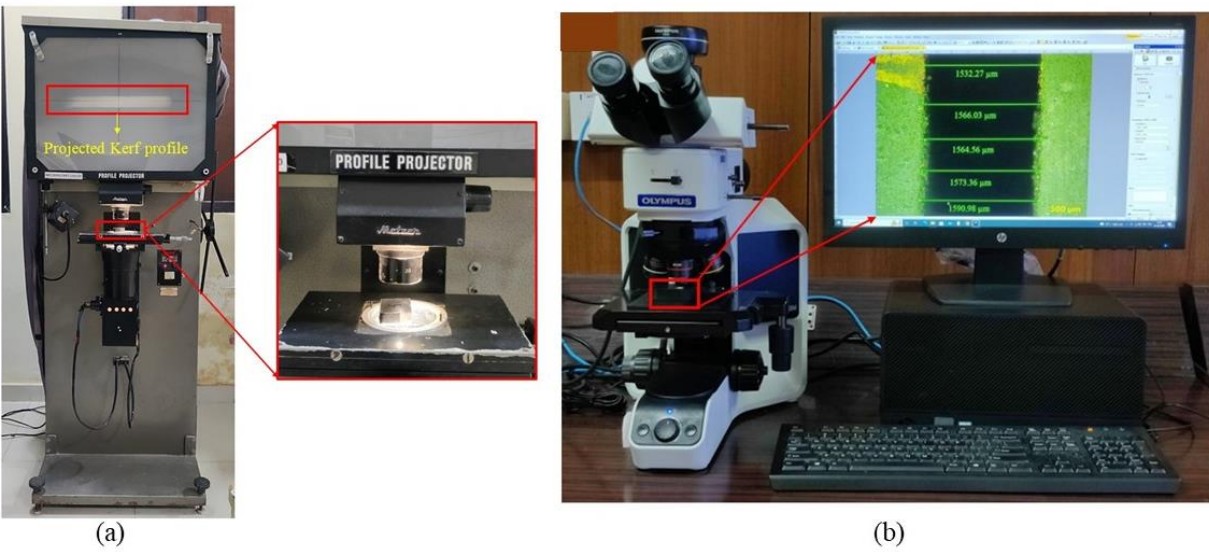

**Figure 5.** Kerf profile measurement using (**a**) profile projector and (**b**) optical microscope.

## 3. Results and Discussions

### 3.1. SEM and EDX Results of Garnet and ABR

The SEM images of the garnet abrasive particles (refer to Figure 6a) were analyzed using ImageJ software to determine the shape factor calculated via Equation (1). For the image of the garnet particles shown in Figure 6a, the procedure for evaluating the shape factor of one sample particle is illustrated in Figure 6b. The shape factor for all the particles was evaluated similarly and subjected to statistical analysis, the results of which are expressed in Figure 7 and Table 3. Figure 7 indicates the distribution of the percentage quantity of the garnet abrasive particles in the sample image. It is visible from the figure that 84% of the garnet abrasive particles had a shape factor of 0.4 to 0.6, whereas 8% of the samples had a shape factor of 0.7 to 0.8. From Table 3, it is observed that the mean and standard deviation of shape factors are 0.5 and 0.12, respectively. The observations from Figure 7 and Table 3 imply that a large proportion of the abrasive particles were sharp edged and aided in the material removal during the machining of ABR in micro-cutting mode.

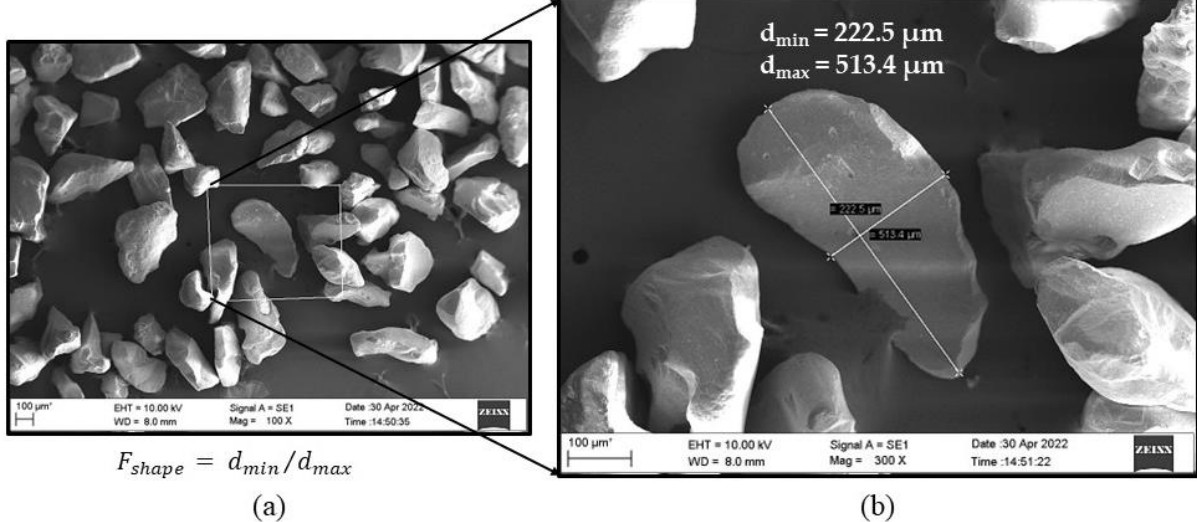

**Figure 6.** Sample evaluation of shape factor for garnet abrasive particles using SEM at (**a**) 100× and (**b**) 300×.

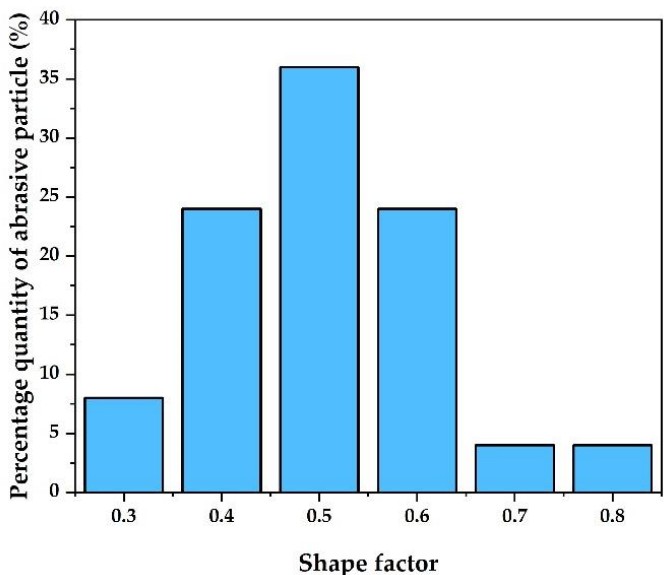

**Figure 7.** Bar chart of percentage quantity of the Shape factor of garnet abrasive particles.

**Table 3.** Shape factors of the garnet abrasive particles visible in the SEM image.

| Particle Number (Pn) | Shape Factor |
| --- | --- |
| Pn1 | 0.3 |
| Pn2 | 0.5 |
| Pn3 | 0.4 |
| Pn4 | 0.7 |
| Pn5 | 0.3 |
| Pn6 | 0.4 |
| Pn7 | 0.6 |
| Pn8 | 0.6 |
| Pn9 | 0.4 |
| Pn10 | 0.4 |
| Pn11 | 0.6 |
| Pn12 | 0.6 |
| Pn13 | 0.4 |
| Pn14 | 0.6 |
| Pn15 | 0.5 |
| Pn16 | 0.6 |
| Pn17 | 0.5 |
| Pn18 | 0.6 |
| Pn19 | 0.5 |
| Pn20 | 0.4 |
| Pn21 | 0.5 |
| Pn22 | 0.5 |
| Pn23 | 0.8 |
| Pn24 | 0.5 |
| Pn25 | 0.6 |
| **Mean** | **0.5** |
| **Standard Deviation** | **0.12** |

The weight percentage of mineral composition present in garnet abrasive particles and ABR were confirmed using SEM-EDX spectroscopy analysis, as shown in Figures 8 and 9, respectively. The study was carried out under a secondary electron mode with an acceleration voltage of 10 kV. The garnet abrasive consisted of elements such as O, Mg, Al, Si, Ca, Ti, Mn, and Fe. The base ABR consisted of elements such as O, Mg, Al, Si, Fe, Cl, Ca, Fe, and Zn.

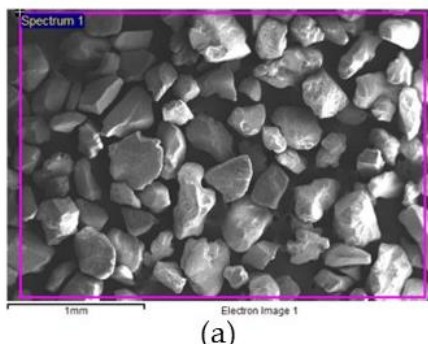
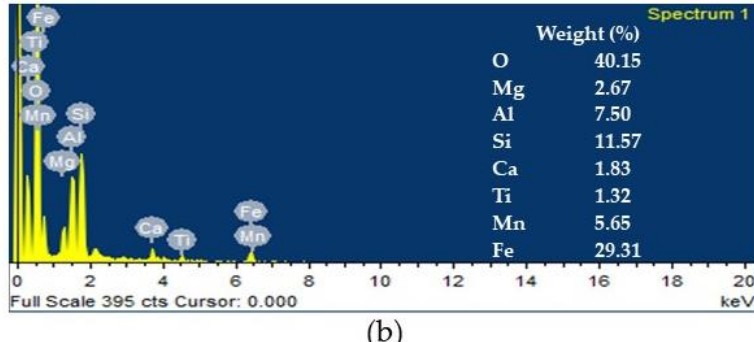

(a)　　　　　　　　　　　　　　　　　　　　　　(b)

**Figure 8.** Microscopic view (**a**) and mineral composition (**b**) of garnet abrasive particles using SEM-EDX.

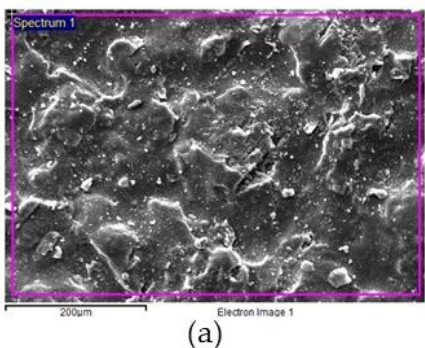
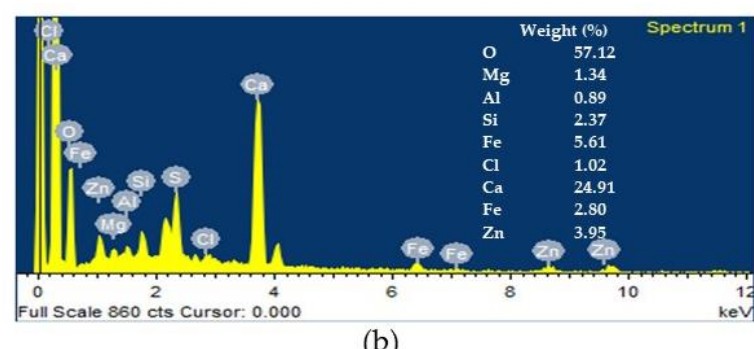

(a)　　　　　　　　　　　　　　　　　　　　　　(b)

**Figure 9.** Microscopic view (**a**) and mineral composition (**b**) of ABR using SEM-EDX.

### *3.2. Effect of Cryogenic Cooling on Kerf Taper Ratio and Material Removal Rate in Slot Machining of ABR*

The $KT_R$ values under conventional and cryogenic conditions are presented in Table 4, which shows the percentage reduction in the $KT_R$ owing to the cryogenic conditions compared to the conventional S-AWJ machining of ABR for various combinations of *WJP*, $V_f$, and *SOD*. The variation in the $KT_R$ for each experimental run under conventional and cryogenic conditions is shown in Figure 10a. The lowest $KT_R$ (1.25) was achieved with a *WJP* of 200 bar, $V_f$ of 40 mm/min, and *SOD* of 1 mm under the cryogenic condition. Figure 10 (b) shows the percentage reduction in $KT_R$ in cryogenic over conventional S-AWJ machining with the mean percentage reduction being 19%. Thus, the $KT_R$ decreases under cryogenic conditions and may be attributed to the $KW_T$ and $KW_B$ being more susceptible to the machining wear mode effects, resulting in a uniform kerf width. The machining kinetic energy of the abrasive particles is maintained throughout the process due to the decreased particle embedding and fragmented abrasive particles. A similar trend was reported by Natrajan et al. [35] during AWJ machining of aluminum alloy.

The percentage increase in the *MRR* under cryogenic conditions compared to the conventional S-AWJ machining of ABR is shown in Table 5. The variation in the *MRR* values for each experimental run under conventional and cryogenic conditions is shown in Figure 11a. Figure 11b shows the percentage improvement in *MRR* in cryogenic over conventional S-AWJ machining, with the mean percentage improvement being 3%. Thus, the *MRR* increases under cryogenic conditions and may be attributed to the sudden increase in Young's modulus and decrease in elasticity of the machining zone resulting in erosion of fine debris (micro-cutting) instead of chip-like debris (microchips) prevalent in conventional AWJ machining [35,36]. The presence of the garnet abrasives further enhanced the participation of micro-cutting activity in the machining zone leading to a higher erosion rate and subsequently increased *MRR*. The highest *MRR* achieved at the *WJP* of 250 bar, $V_f$ of 60 mm/min, and *SOD* of 2.0 mm under cryogenic conditions is 1604.84 mm$^3$/min.

**Table 4.** Variations in the $KT_R$ in the S-AWJ machining of ABR under conventional and cryogenic conditions.

| Experiment No. | WJP | $V_f$ | SOD | $KT_R$ | | % Reduction of $KT_R$ in Cryogenic over Conventional S-AWJ Machining |
| --- | --- | --- | --- | --- | --- | --- |
| | | | | Conventional | Cryogenic | |
| EN1 | 150 | 40 | 1.0 | 1.79 | 1.44 | 19.56 |
| EN2 | 150 | 40 | 1.5 | 1.71 | 1.48 | 13.46 |
| EN3 | 150 | 40 | 2.0 | 1.69 | 1.52 | 10.06 |
| EN4 | 150 | 50 | 1.0 | 2.20 | 1.77 | 19.55 |
| EN5 | 150 | 50 | 1.5 | 2.21 | 1.80 | 18.56 |
| EN6 | 150 | 50 | 2.0 | 2.28 | 1.82 | 20.18 |
| EN7 | 150 | 60 | 1.0 | 2.13 | 2.23 | −4.70 |
| EN8 | 150 | 60 | 1.5 | 2.28 | 2.24 | 1.76 |
| EN9 | 150 | 60 | 2.0 | 2.54 | 2.26 | 11.03 |
| EN10 | 200 | 40 | 1.0 | 1.57 | 1.25 | 20.39 |
| EN11 | 200 | 40 | 1.5 | 1.47 | 1.29 | 12.25 |
| EN12 | 200 | 40 | 2.0 | 1.42 | 1.30 | 8.46 |
| EN13 | 200 | 50 | 1.0 | 1.92 | 1.39 | 27.61 |
| EN14 | 200 | 50 | 1.5 | 1.87 | 1.42 | 24.07 |
| EN15 | 200 | 50 | 2.0 | 1.86 | 1.44 | 22.59 |
| EN16 | 200 | 60 | 1.0 | 1.95 | 1.57 | 19.49 |
| EN17 | 200 | 60 | 1.5 | 2.00 | 1.60 | 20.00 |
| EN18 | 200 | 60 | 2.0 | 2.11 | 1.62 | 23.23 |
| EN19 | 250 | 40 | 1.0 | 1.72 | 1.37 | 20.35 |
| EN20 | 250 | 40 | 1.5 | 1.56 | 1.38 | 11.54 |
| EN21 | 250 | 40 | 2.0 | 1.46 | 1.38 | 5.48 |
| EN22 | 250 | 50 | 1.0 | 2.15 | 1.44 | 33.03 |
| EN23 | 250 | 50 | 1.5 | 2.03 | 1.45 | 28.58 |
| EN24 | 250 | 50 | 2.0 | 1.95 | 1.45 | 25.65 |
| EN25 | 250 | 60 | 1.0 | 2.24 | 1.53 | 31.70 |
| EN26 | 250 | 60 | 1.5 | 2.24 | 1.55 | 30.81 |
| EN27 | 250 | 60 | 2.0 | 2.29 | 1.55 | 32.32 |

**Table 5.** Variations in the *MRR* in the S-AWJ machining of ABR under conventional and cryogenic conditions.

| Experiment No. | WJP | $V_f$ | SOD | MRR | | % Improvement of *MRR* in Cryogenic over Conventional S-AWJ Machining |
| --- | --- | --- | --- | --- | --- | --- |
| | | | | Conventional | Cryogenic | |
| EN1 | 150 | 40 | 1.0 | 795.12 | 801.09 | 0.75 |
| EN2 | 150 | 40 | 1.5 | 856.80 | 884.16 | 3.10 |
| EN3 | 150 | 40 | 2.0 | 918.35 | 947.82 | 3.11 |
| EN4 | 150 | 50 | 1.0 | 995.72 | 934.95 | −6.50 |
| EN5 | 150 | 50 | 1.5 | 1018.47 | 1025.55 | 0.70 |
| EN6 | 150 | 50 | 2.0 | 1041.05 | 1091.85 | 4.66 |
| EN7 | 150 | 60 | 1.0 | 1346.15 | 1132.43 | −18.88 |
| EN8 | 150 | 60 | 1.5 | 1308.21 | 1225.17 | −6.78 |
| EN9 | 150 | 60 | 2.0 | 1270.07 | 1288.80 | 1.46 |
| EN10 | 200 | 40 | 1.0 | 926.09 | 969.90 | 4.52 |
| EN11 | 200 | 40 | 1.5 | 973.92 | 1035.51 | 5.95 |
| EN12 | 200 | 40 | 2.0 | 1021.61 | 1081.68 | 5.56 |
| EN13 | 200 | 50 | 1.0 | 1141.22 | 1140.57 | −0.06 |
| EN14 | 200 | 50 | 1.5 | 1146.65 | 1209.30 | 5.19 |
| EN15 | 200 | 50 | 2.0 | 1151.90 | 1253.78 | 8.13 |
| EN16 | 200 | 60 | 1.0 | 1498.89 | 1372.55 | −9.21 |
| EN17 | 200 | 60 | 1.5 | 1440.17 | 1439.10 | −0.08 |
| EN18 | 200 | 60 | 2.0 | 1381.24 | 1476.54 | 6.46 |
| EN19 | 250 | 40 | 1.0 | 1018.87 | 1099.11 | 7.31 |
| EN20 | 250 | 40 | 1.5 | 1052.85 | 1147.20 | 8.23 |
| EN21 | 250 | 40 | 2.0 | 1086.68 | 1175.97 | 7.60 |
| EN22 | 250 | 50 | 1.0 | 1238.99 | 1296.57 | 4.45 |
| EN23 | 250 | 50 | 1.5 | 1227.10 | 1343.48 | 8.67 |

**Table 5.** *Cont.*

| Experiment No. | WJP | $V_f$ | SOD | MRR | | % Improvement of *MRR* in Cryogenic over Conventional S-AWJ Machining |
|---|---|---|---|---|---|---|
| | | | | Conventional | Cryogenic | |
| EN24 | 250 | 50 | 2.0 | 1215.03 | 1366.13 | 11.07 |
| EN25 | 250 | 60 | 1.0 | 1594.36 | 1553.27 | −2.65 |
| EN26 | 250 | 60 | 1.5 | 1514.85 | 1593.63 | 4.95 |
| EN27 | 250 | 60 | 2.0 | 1435.14 | 1604.84 | 10.58 |

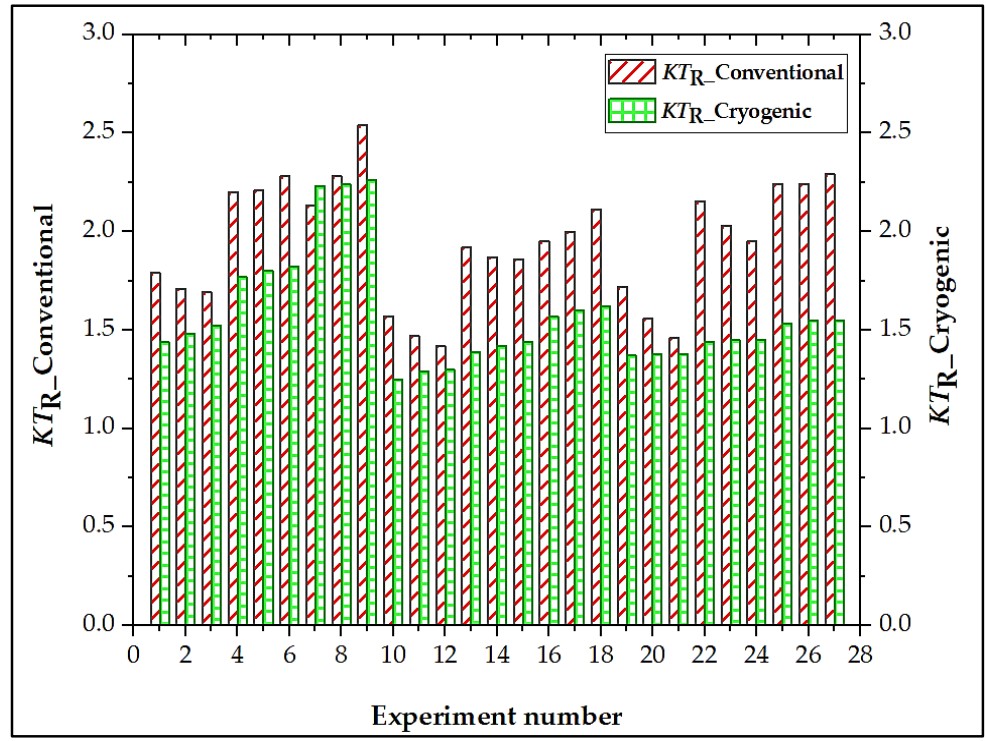

(a)

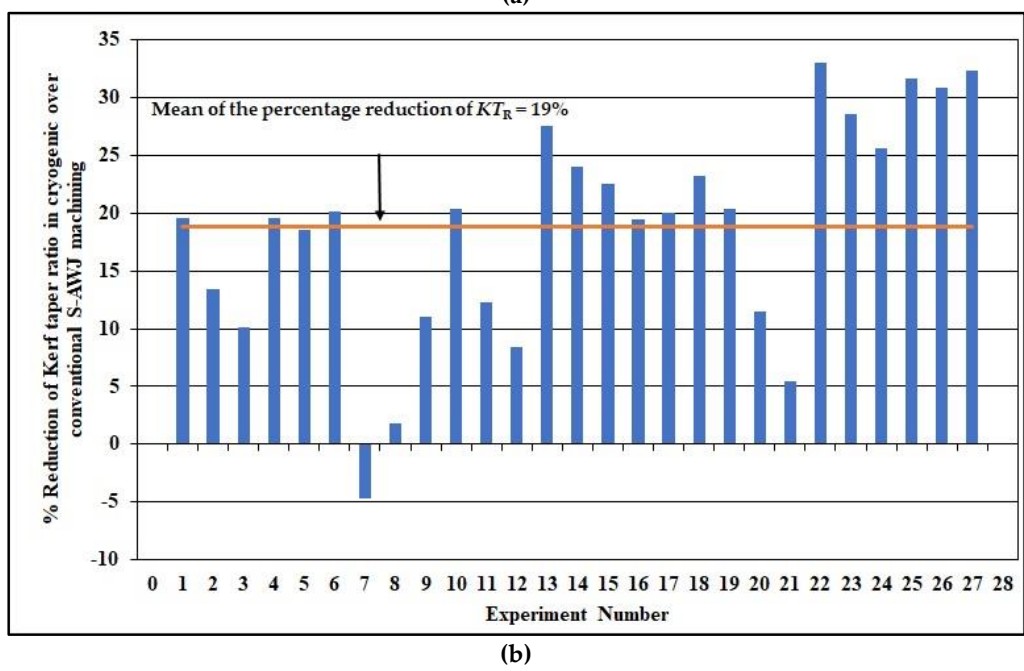

(b)

**Figure 10.** (**a**) Experimental $KT_R$ under conventional and cryogenic conditions and (**b**) percentage reduction of $KT_R$ in cryogenic over conventional S-AWJ machining.

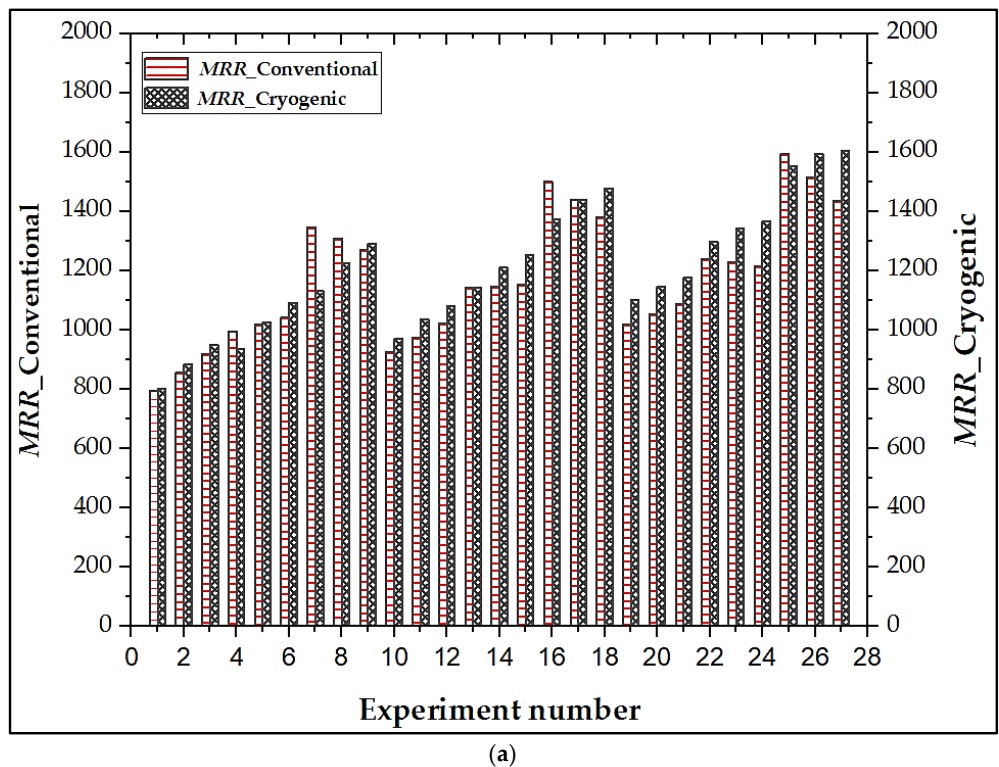

(**a**)

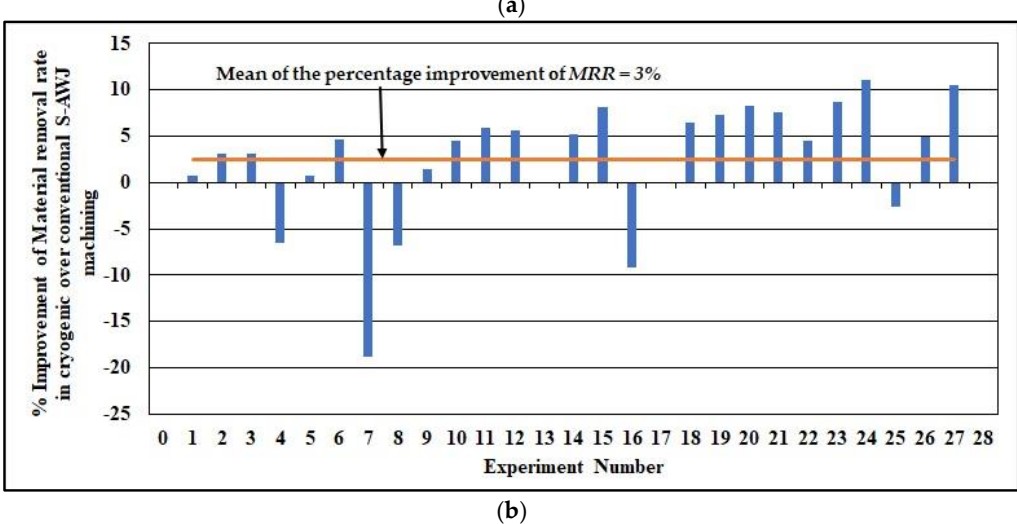

(**b**)

**Figure 11.** (**a**) Experimental *MRR* under conventional and cryogenic conditions and (**b**) percentage improvement of material removal rate in cryogenic over conventional S-AWJ machining.

### 3.3. Effect of Cryogenic Cooling on Kerf Profile Produced during Slot Machining of the ABR Material

The top and bottom kerf profile images of all experimental test sequences, obtained from the profile projector and optical microscope, are presented in Figures 12 and 13, respectively. The top kerf profiles of the machined slots are of good quality for both conditions (conventional and cryogenic). The upcoming sub-section includes the machined kerf profile details at all experimental test sequences obtained at the *WJP* of 150 bar, 200 bar and 250 bar under conventional and cryogenic conditions.

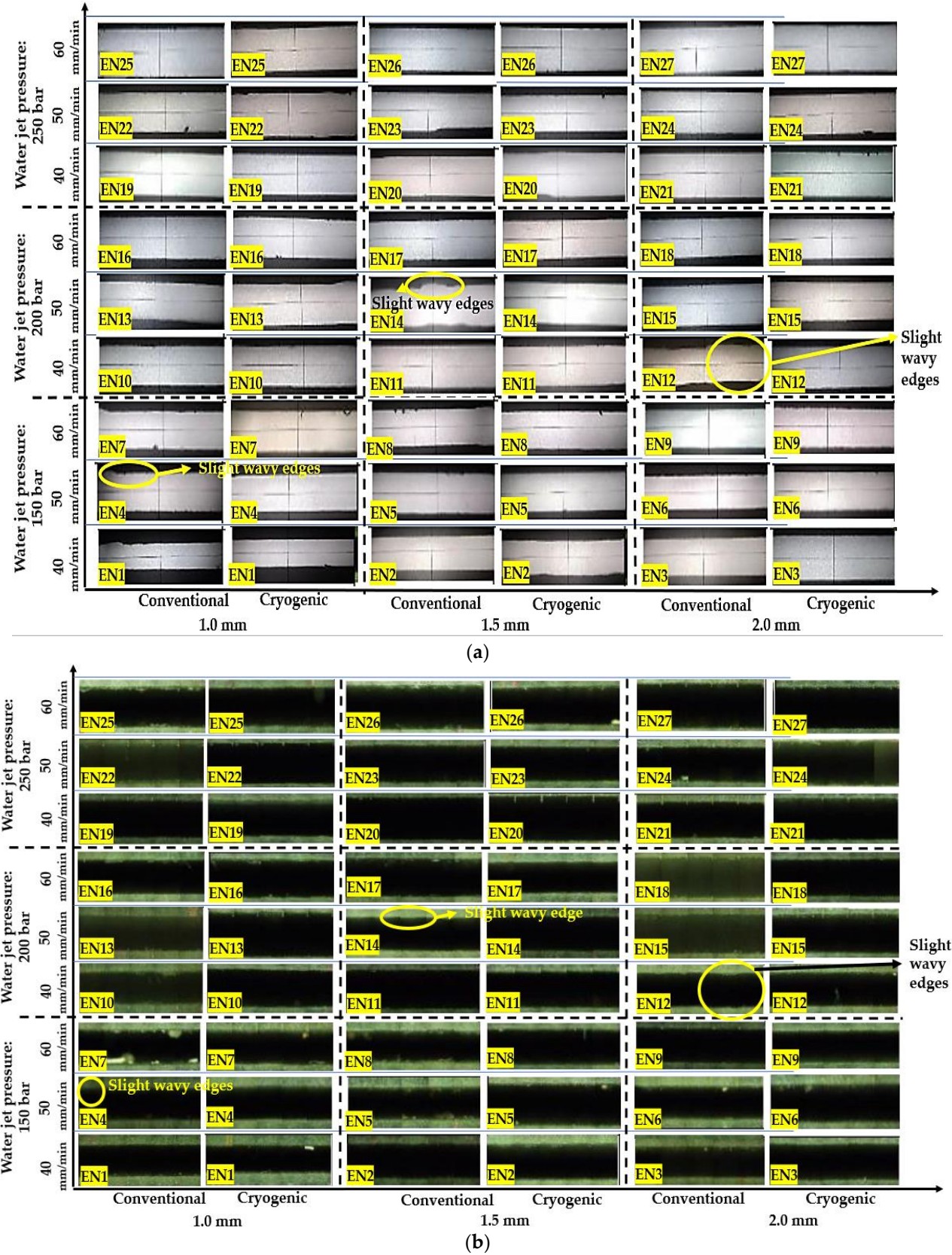

**Figure 12.** Top kerf profile images of machined slots on ABR, obtained from the S-AWJ machining test sequences under conventional and cryogenic conditions, using (**a**) profile projector and (**b**) optical microscope.

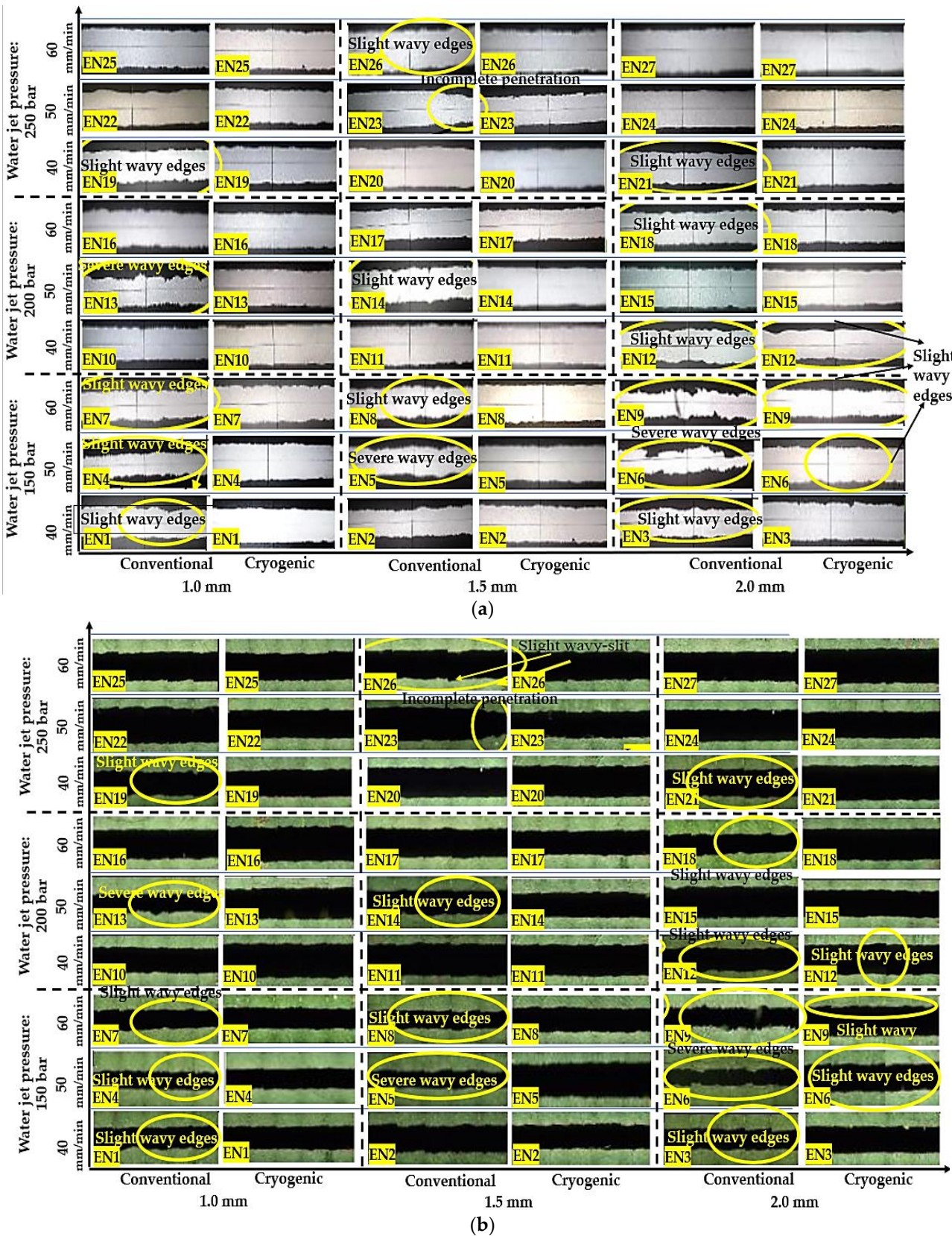

**Figure 13.** Bottom kerf profile images of machined slots on ABR, obtained from the S-AWJ machining test sequences under conventional and cryogenic conditions, using (**a**) profile projector and (**b**) optical microscope.

### 3.3.1. At WJP 150 Bar

The *WJP* at 150 bar produced slight and severe wavy edges at the bottom kerf profiles under conventional conditions. The experimental sequence with *WJP* of 150 bar and *SOD* of 2 mm with increased $V_f$ produced severe wavy edges at the bottom kerf profiles leading to raised $KT_R$ (2.54, refer to Table 4: experiment number: 9) despite the decreased $KW_T$ and $KW_B$. This can be attributed to the lag produced due to fast nozzle movement combined with high energy attenuation of the downstream jet. This is caused by the jet remaining in the unit area of the workpiece for a shorter duration resulting in lower material removal with greater $V_f$. When *SOD* is higher, there is more space between the nozzle and the workpiece's top surface, and the abrasive particles take more time to accelerate before striking the workpiece. Furthermore, the $KW_T$ also increased due to the concurrent rise in the cross-sectional area of the suspension jet in contact with the workpiece. Conversely, the energy density and abrasive concentration dropped, whereas the downstream jet diverged more, leading to increased $KT_R$ and wavy edges. This is visible in experiment numbers 3, 6, and 9, as shown in Figure 13.

On the other hand, the machining of ABR under cryogenic conditions produced an improved bottom kerf profile in most of the experimental sequences. This can be attributed to the strengthening of the molecular chains of ABR under cryogenic conditions and the slow inhibition of the elastic property of ABR due to a decrease in the interatomic distance between molecules. It ultimately resulted in ductile erosion, thereby preventing the formation of wavy edges, but slight wavy edges were observed at low *WJP* (150 bar) due to the decreased kinetic energy of the jet at the bottom of the machined surface.

### 3.3.2. At *WJP* 200 Bar

The *WJP* at 200 bar with increased $V_f$ and *SOD* produced slight and severe wavy edges at the bottom kerf profiles under conventional condition, as shown in Figure 13a,b. The energy density and abrasive concentration dropped with the increased *SOD*, and the downstream jet diverged more, which is the primary reason for increased $KT_R$ and severe wavy edges.

On the other hand, the machining of ABR under cryogenic conditions at higher *WJP* (200 bar) produced waviness-free profiles at the top and bottom kerf surfaces in most of the experimental sequences. This is possibly due to the transformation of ABR from its elastic to tough phase, which is attributed to the enhanced Young's modulus and machinability properties. Additionally, the high *WJP* caused increased kinetic energy and amount of jet diffusion of the abrasive particles to produce the satisfying quality of the cut profiles.

### 3.3.3. At *WJP* 250 Bar

The machining ability and the amount of jet diffusion were improved with an increase in the *WJP* (250 bar) under conventional conditions. As a result, the *MRR* improved, and only slightly wavy edges occurred at increased $V_f$ and *SOD*. The jet ability of the downstream to machine effectively and remove material at a faster rate was made possible by the high *WJP* and $V_f$. This is the prime reason for reduced waviness, as observed in the experimental sequences from 19 through to 27 (refer Figure 13a,b). Kalla et al. [37] and Fowler et al. [38] found similar behavior in the S-AWJ machining of Graphite epoxy laminates and Titanium alloy, respectively.

Conversely, the machined profiles under cryogenic conditions at the highest *WJP* (250 bar) produced the waviness-free top and bottom kerf profiles in most of the experimental sequences from 19 through to 27 (refer Figure 13a,b). This is attributed to the possible ductile erosion.

### 3.4. Effect of Cryogenic Conditions on the Abrasive Contamination during the AWJ Machining of ABR

Earlier research has demonstrated that the use of $LN_2$ significantly reduces the amount of particle embedding in the elastomeric polymer, i.e., polydimethylsiloxane and acry-

lonitrile butadiene styrene and polytetrafluoroethylene, etc., during abrasive jet machining [39–41]. A relative evaluation of abrasive particle embedding rather than an absolute measurement was conducted in [39–41]. In the current work, the S-AWJ-machined ABR workpieces under conventional and cryogenic conditions were prepared and analyzed for SEM and EDX analysis, as described in Section 2.3. Fixed S-AWJ machining process parameters were selected to compare the abrasive contamination of ABR under conventional and cryogenic conditions. The preferred process parameters were 250 bar of *WJP*, 50 mm/min of $V_f$ and 1.5 mm of *SOD*. The SEM images and EDX spectrum of the top kerf wall of the machined ABR workpiece under conventional and cryogenic conditions are shown in Figure 14a,b, respectively. The analysis indicates the abrasive chemical element present in the cut surface. The compound $SiO_2$ is disintegrated into Silicon (Si) and Oxygen ($O_2$) elements during the S-AWJ machining process. Hence, Si was embedded at the top surface of the machined slot during conventional and cryogenic conditions, leading to an increased weight percentage of Si particles present on the base material of ABR. Along with Si, Manganese (Mn) was also embedded at the top surface of the machined slot during conventional and cryogenic conditions. The presence of Mn as an embedded particle is considered "abrasive particle contamination" since it was not present in the ABR base material composition (refer to Figure 9). The weight percentage of Si and Mn particles present in the S-AWJ and cryogenic-assisted S-AWJ machining conditions are shown in Figure 15. This confirms that the weight percentage of Si and Mn particle contamination is lower at the top surface of the machined slot of ABR under cryogenic conditions, compared to that in the conventional conditions. The reduced Si and Mn particle contamination can be attributed to the increased elastic modulus of ABR and change in the erosion process under cryogenic condition.

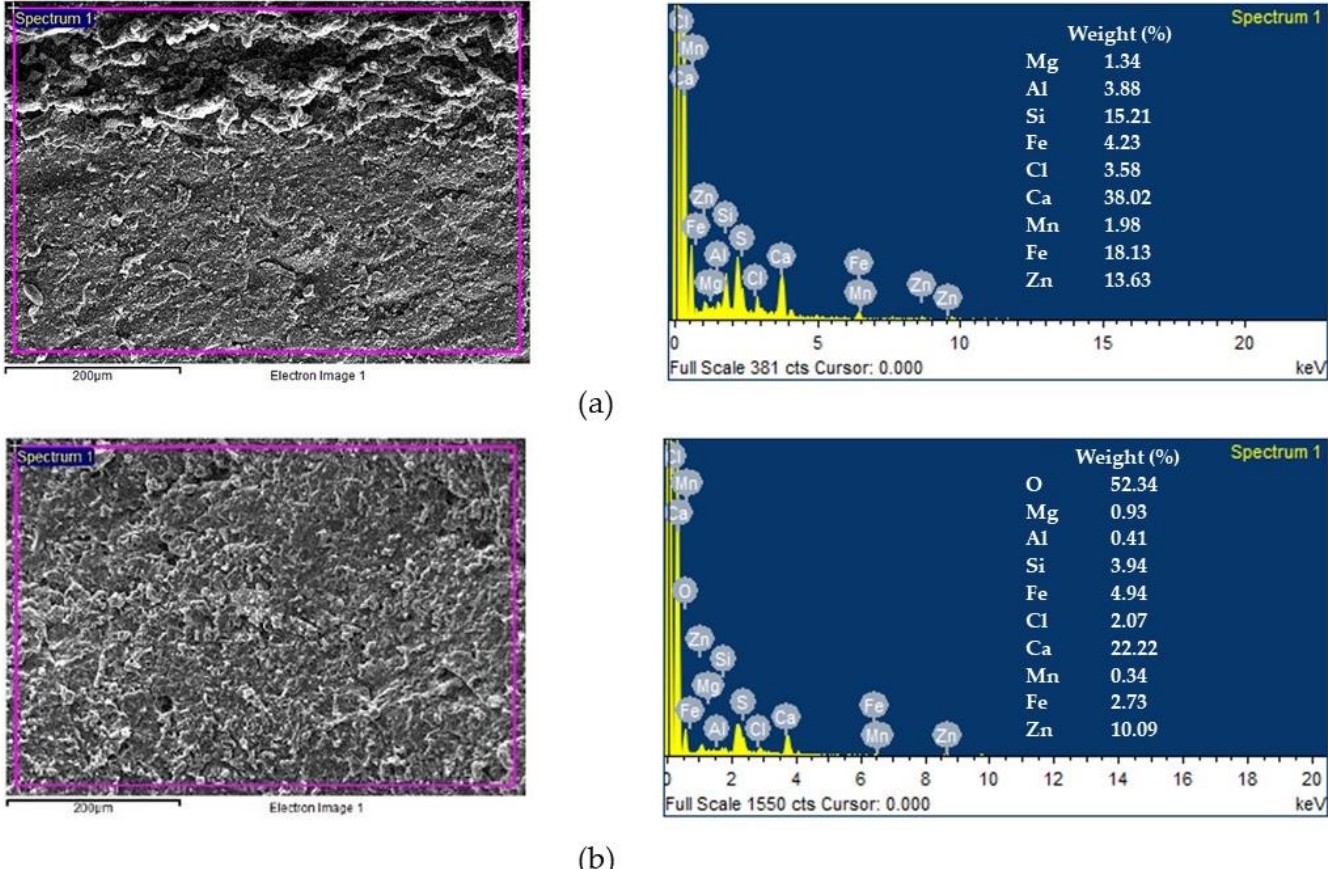

**Figure 14.** SEM image and EDX spectra of abrasive contamination at the top kerf wall of machined ABR workpiece under (**a**) conventional and (**b**) cryogenic conditions.

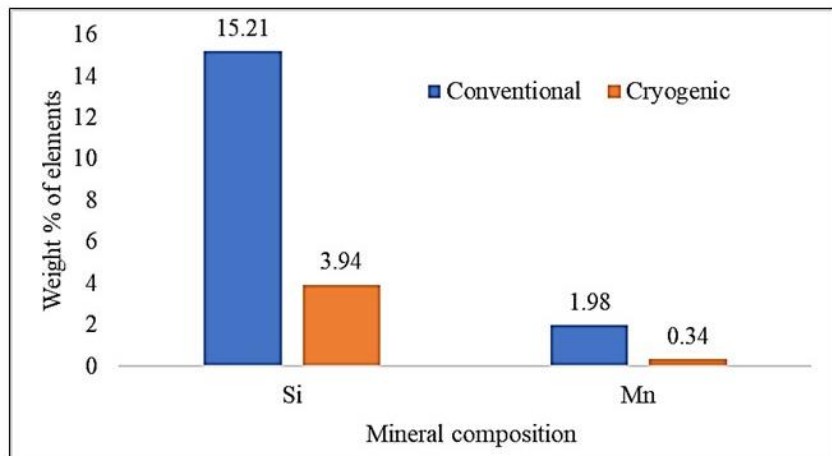

**Figure 15.** Weight percentage of abrasive particle contamination during the S-AWJ machining of ABR under conventional and cryogenic conditions.

The visual comparison of abrasive particle embedding on the machined ABR workpiece under conventional and cryogenic conditions was made by taking the SEM images in BSE mode. The similar process parameter conditions (250 bar of *WJP*, 50 mm/min of $V_f$, and 1.5 mm of *SOD*) under conventional and cryogenic were compared for SEM-EDX analysis. The white dots in the BSE micrograph of the machined ABR surface under the conventional and cryogenic conditions, as shown in Figure 16a and b, respectively, represent the "abrasive particle contamination". Using ImageJ, the BSE images were reversed and then filtered to remove noise using a medium filter with a 2-pixel radius. In Figure 17a,b, a 1 pixel × 1 pixel area presented 1 $\mu m^2$, and a lower particle detection threshold of 10 $\mu m^2$ was used. Comparing Figure 17a,b, it is clear that using $LN_2$ reduced the number of embedded garnet particles at the machined ABR surface significantly. In conclusion, the embrittlement of ABR caused by $LN_2$ limits the embedding of garnet particles and enhances surface quality. These outcomes are in line with the findings of the previous researchers during cryogenic abrasive jet machining of acrylonitrile butadiene styrene [39], polytetrafluoroethylene [40], and polydimethylsiloxane [41].

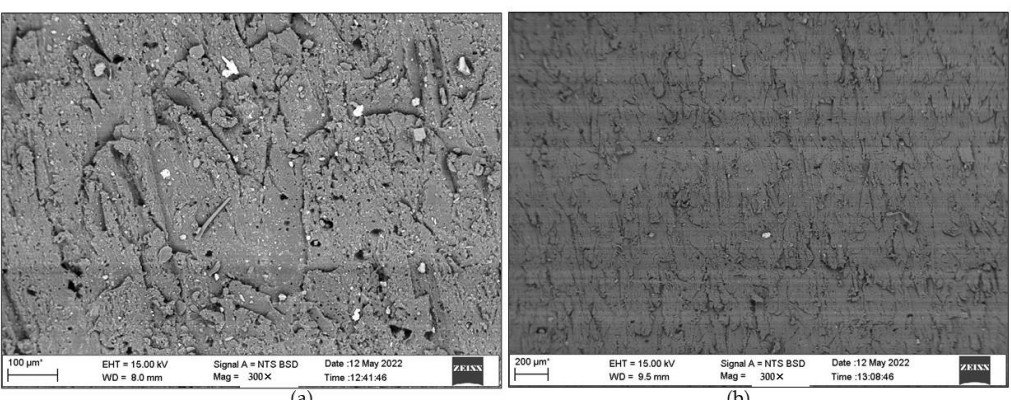

**Figure 16.** Backscatter scanning electron micrograph of garnet particles embedded in the machined surface of ABR at *WJP* of 250 bar, *V*f of 50 mm/min and *SOD* of 1.5 mm/min under the (**a**) conventional and (**b**) cryogenic conditions.

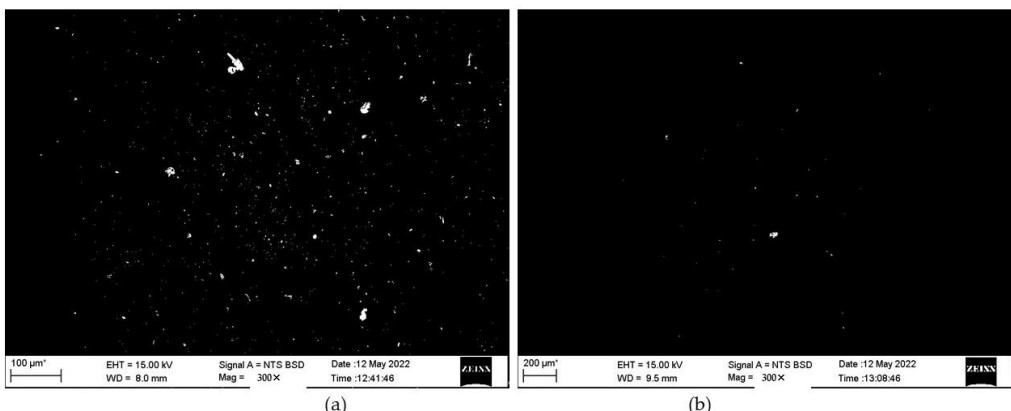

**Figure 17.** Images obtained using ImageJ software under the (**a**) conventional and (**b**) cryogenic conditions; white dots represent garnet particle embedding.

## 4. Conclusions

In the current work, the S-AWJ machining is carried out on the ABR under conventional and cryogenic conditions. The effect of cryogenic conditions on the $KT_R$ and *MRR* is analyzed. The kerf characteristics study was conducted at the top surface near the vicinity of the machined slot. The machined ABR workpieces' kerf characteristics and surface quality were compared under conventional and cryogenic conditions. The following conclusions are drawn from the conducted study.

- The lowest $KT_R$ (1.25) was achieved under cryogenic conditions of *WJP* of 200 bar, $V_f$ of 40 mm/min, and *SOD* of 1.0 mm.
- The highest *MRR* (1604.84 mm$^3$/min) was achieved under cryogenic conditions of *WJP* of 250 bar, $V_f$ of 60 mm/min, and *SOD* of 2.0 mm.
- The cryogenic conditions improved the machined cut profile to a large extent.
- The reduced waviness during the cryogenic-assisted S-AWJ machining showed better uniformity in the geometry of the cut slot compared to conventional conditions due to increased Young's modulus and decreased elasticity of ABR.
- Under conventional conditions, particularly with low *WJP*, high *SOD*, and $V_f$ levels, severe and slight wavy edges could be seen at the bottom kerf profiles.
- Slight wavy edges were also observed in the bottom kerf profiles produced at low *WJP* under cryogenic conditions.
- The use of LN$_2$ during S-AWJ machining of ABR resulted in the decreased percentage of Si and Mn particles on the top kerf machined profile due to the increased elastic modulus of ABR and the change in the erosion process.
- The cryogenic-assisted S-AWJ-machined ABR workpiece showed enhanced surface quality.

From the investigation of the research work presented in this paper, it can be concluded that cryogenic-assisted S-AWJ machining of ABR exhibited improved machining quality in terms of lower $KT_R$ and higher *MRR*. Better kerf characteristics were achieved through uniform and waviness-free kerf profiles as well as lower abrasive particle embedding in the machined surfaces. Further investigation is recommended by considering a wide range of machining parameters and a large number of machined slots to enhance the performance and kerf characteristics during the machining of ABR workpieces.

**Author Contributions:** P.M., conceptualization, investigation, writing—original draft; G.S.V., methodology, data curation, supervision, project administration, funding acquisition; R.C.K., conceptualization, writing—review and editing, formal analysis, visualization, writing—original draft preparation. All authors have read and agreed to the published version of the manuscript.

**Funding:** The current research received the Intra mural fund from the Manipal Academy of Higher Education (MAHE/DREG/PhD/IMF/2019).

**Data Availability Statement:** Not applicable.

**Acknowledgments:** The authors thank the Manipal Academy of Higher Education (MAHE), Manipal for providing administrative and technical support. The authors acknowledge the statistical analysis service provided by Sumathi K., Department of Mathematics, Manipal Institute of Technology (MIT), Manipal.

**Conflicts of Interest:** The authors declare no conflict of interest.

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
