# Peer review of "Investigation on Performance and Kerf Characteristics during Cryogenic-Assisted Suspension-Type Abrasive Water Jet Machining of Acrylonitrile Butadiene Rubber"

_jcs, doi:10.3390/jcs6120397_

Round 1
Reviewer 1 Report
The paper entitled " Investigation on Performance and Kerf Characteristics during Cryogenic-assisted Suspension-type Abrasive Water Jet Machining of Acrylonitrile Butadiene Rubber" aims to compare the effect of cryogenic condition on the performance parameters of the S-AWJ machining and investigate the kerf characteristics of the top and bottom surface. From my point of view, the topic is of great interest, but the overall quality should be improved:
· A graphical abstract would add interest to catch the eye
· The introduction seems to be correct and gives a good review of the literature.
· Table 3 could be summarised as a 3-level experiment with 3 factors. The summarised version of the table I think would be more than sufficient.
· Repetitions of the machining conditions are not envisaged. It would be interesting to know what the pure error of the experiments is.
· The order of trials is also not reflected. If this has not been done, consider randomising the trials. The evolution from one condition to the other may affect some trials.
· Figure 10 and Figure 11 is difficult to follow seems a little small.
I liked the conclusions, especially the future direction.
Reviewer 2 Report
In this study, the authors reported a cryogenic-assisted suspension-type abrasive water jet machining of ABR and studied the performance and kerf characteristics. They found that compared to the conventional method, the new method positively affected the performance parameters. These results are important and helpful in the relevant applications. In general, the work is robust and the mechanism is quite clear, although some non-fatal problems still remain. Therefore, the reviewer believe the manuscript can be accepted after a moderate revision by properly addressing the following questions:
1. The determination of the shape factor should base on statistics of at least several particles. The associated standard deviation should be presented to analyze the reliability of the results.
2. Table 3 is fully included by Table 4 and hence should be removed from the manuscript.
3. Why the image size is so different for Figure 6 and Figure 7?
4. It is not very clear to see that MRR under cryogenic conditions is remarkably higher than that under conventional conditions. To show that clearly, one can plot the %improvement (y-axis) against the experimental number (x-axis), as a sub-figure, to see whether the line is in the whole higher than the zero line. This should be also applied to Figure 8.
5. Is there any evidence that the Young’s modulus suddenly increases during the cryogenic condition in this study? Although the modulus usually increases with the decreasing temperature, it is still relying on the freezing time, the bulk temperature and etc, which seems still a speculation here.
6. Figure 12, why the weight of Cl element is rather low, while the peak is very high?
7. The serial number of all the figures are wrong – there are two “Figure 1” in the manuscript.
8. Some of the figures have a very low resolution, which is hard for the readers and reviewers to see the detail, e.g., Figure 1 (the one reproduce with permission), Figure 6a, Figure 12b and etc.
Round 2
Reviewer 1 Report
I see no impediment to the publication of this paper.
Reviewer 2 Report
All of my questions have been solved properly by the authors. I appreciate the authors for their efforts, and I believe the manuscript can be accepted as it is.